# Reliable Flow-Cytometric Approach for Minimal Residual Disease Monitoring in Patients with B-Cell Precursor Acute Lymphoblastic Leukemia after CD19-Targeted Therapy

**DOI:** 10.3390/cancers14215445

**Published:** 2022-11-05

**Authors:** Ekaterina Mikhailova, Olga Illarionova, Alexander Komkov, Elena Zerkalenkova, Ilgar Mamedov, Larisa Shelikhova, Yulia Olshanskaya, Natalia Miakova, Galina Novichkova, Alexander Karachunskiy, Michael Maschan, Alexander Popov

**Affiliations:** 1Dmitry Rogachev National Medical Research Center of Pediatric Hematology, Oncology and Immunology, 117998 Moscow, Russia; 2Department of Genomics of Adaptive Immunity, Shemyakin-Ovchinnikov Institute of Bioorganic Chemistry, 117998 Moscow, Russia

**Keywords:** minimal residual disease, flow cytometry, blinatumomab, CAR-T cells

## Abstract

**Simple Summary:**

We aimed to develop an antibody panel and data analysis algorithm for multicolor flow cytometry (MFC), which is a reliable method for minimal residual disease (MRD) detection in patients with B-cell precursor acute lymphoblastic leukemia (BCP-ALL) treated with CD19-directed therapy. We have developed a single-tube 11-color panel for MFC-MRD detection, which was adapted for the case of possible CD19 loss. Based on patterns of antigen expression changes and the relative expansion of normal CD19-negative BCPs, guidelines for MFC data analysis and interpretation were established. The suggested approach was tested in comparison with the molecular techniques with a high rate of qualitative concordance obtained.

**Abstract:**

We aimed to develop an antibody panel and data analysis algorithm for multicolor flow cytometry (MFC), which is a reliable method for minimal residual disease (MRD) detection in patients with B-cell precursor acute lymphoblastic leukemia (BCP-ALL) treated with CD19-directed therapy. The development of the approach, which was adapted for the case of possible CD19 loss, was based on the additional B-lineage marker expression data obtained from a study of primary BCP-ALL patients, an analysis of the immunophenotypic changes that occur during blinatumomab or CAR-T therapy, and an analysis of very early CD19-negative normal BCPs. We have developed a single-tube 11-color panel for MFC-MRD detection. CD22- and iCD79a-based primary B-lineage gating (preferably consecutive) was recommended. Based on patterns of antigen expression changes and the relative expansion of normal CD19-negative BCPs, guidelines for MFC data analysis and interpretation were established. The suggested approach was tested in comparison with the molecular techniques: *IG/TR* gene rearrangement detection by next-generation sequencing (NGS) and RQ-PCR for fusion-gene transcripts (FGTs). Qualitative concordance rates of 82.8% and 89.8% were obtained for NGS-MRD and FGT-MRD results, respectively. We have developed a sensitive and reliable approach that allows MFC-MRD monitoring after CD19-directed treatment, even in the case of possible CD19 loss.

## 1. Introduction

Multicolor flow cytometry (MFC) is a widely applicable technology for minimal residual disease (MRD) monitoring in patients with B-cell precursor acute lymphoblastic leukemia (BCP-ALL) [1,2]. MFC is relatively faster, less expensive and less laborious than methods used for the detection of fusion genes, their transcripts (FGTs) or patient-specific rearrangements of immunoglobulin and T-cell receptor genes (*IG/TR*), such as either real-time quantitative PCR (RQ-PCR) or next-generation sequencing (NGS) [2,3]. Moreover, MFC allows for the detailed analysis of bone marrow (BM) cell compartments, as well as of the immunophenotype of the residual leukemic population [3]. Last, it is crucial for the selection of targets for immunotherapy [3], which is now frequently applied for slow MRD responders or in patients with MRD reappearance [4,5]. The active use of immunotherapeutic drugs against CD19 (the bispecific T-cell engager blinatumomab and T cells with chimeric antigen receptors (CAR-T)) [6,7] significantly complicates MFC-MRD monitoring because of possible CD19 downregulation by tumor cells [8,9]. This possible CD19 negativity breaks well-established algorithms of MFC data interpretation [10], as CD19 is typically used for primary B-lineage gating with the subsequent analysis of the CD19-positive compartment [11,12]. Moreover, the selective pressure of the CD19-directed treatment increases the incidence of the so-called lineage switch [13,14], which has become one of the possible mechanisms of BCP-ALL resistance to immunotherapy. Therefore, it is necessary to modify the conventional MFC-MRD approach to overcome the mentioned problems.

Currently, there is no consensus approach to MFC-MRD monitoring after blinatumomab or CAR-T therapy for BCP-ALL. There are several described changes in the MFC-MRD methodology, including the addition of other early B-lineage antigens [15,16] or only the modification of the gating algorithm using conventional sets of antibodies [17]. Based on our previously obtained data, we aimed to develop an antibody panel and MFC data analysis algorithm, which is a reliable method for MFC-MRD detection in patients with BCP-ALL treated with a CD19-directed treatment.

## 2. Methods

### 2.1. Prerequisites

As therapy-related changes in antigen expression can hamper the detection of the initially found leukemia-associated immunophenotype (LAIP), such immunophenotypic shifts were studied in children with relapsed/refractory (R/R) BCP-ALL treated with blinatumomab [18] and CAR-T therapy [19]. A loss of CD19 was found in one-fourth of the patients who relapsed or were MFC-MRD-positive after blinatumomab, whereas after CAR-T therapy, this CD19 downregulation was noted in one-half of the patients. Because of the high incidence of CD19 negativity, it was necessary to search for a possible substitution for CD19 as the primary B-lineage antigen with antigen(s) of similar diagnostic value. The initial expression of CD22, CD24, CD10 and intracellular (i) CD79a was tested in a large cohort of primary BCP-ALL pediatric patients (*n* = 519) [20]. It was found that the tested antigens were expressed by nearly all tumor cells (more than 95% of positive cells in the leukemic population) in 88.9% (CD22), 83.7% (iCD79a), 88.2% (CD10) and 94.7% (CD24) of patients [20]. Moreover, at least one of the studied markers was expressed on nearly all blasts in more than 95% of patients, and there were no patients in whom none of these antigens were found. Therefore, it was recommended to add CD22, CD24 and iCD79a to the conventional antibody panel [20]. At the same time, if the earliest B-lineage antigens (CD22, CD10 and iCD79a) are used for primary B-lineage gating, the B-cell compartment will encompass all stages of BCP maturation, not only CD19 positivity [21]. In this case, the early normal CD19-negative BCP will appear in the B-cell gate [22] and can be misinterpreted as the development of a CD19-negative relapse [22]. We have described the expression of MRD-related antigens by these early BCPs and found their relative expansion in R/R BCP-ALL patients treated with either blinatumomab or CAR-T cells [22]. Moreover, in children who received blinatumomab during frontline treatment immediately after induction, normal CD19-negative BCPs in substantial quantities were detected in all patients at all evaluated time points [23]. Thus, the described changes in the normal background inside the B-lineage compartment should also be considered during MFC data analysis and interpretation.

Other antigens, typically used for MFC-MRD studies, also demonstrated expression instability during CD19 targeting [18,19]. CD34 was the most unstable marker, and its up- or downregulation was registered in nearly one-third of patients in both the blinatumomab and CAR-T cohorts. In contrast, CD58 was remarkably stable, as were CD22 and CD24. Other antigens, such as CD10, CD20, CD34, CD38 and CD45, displayed either an increase or decrease in expression in several patients [18,19]; thus, it was realized that relying mainly on the initial LAIP is erroneous. At the same time, we have seen an increased proportion of lineage switching [18,24], mainly as the result of resistance to blinatumomab rather than the mechanism of relapse development [24]. Therefore, possible lineage conversion also must be considered during MFC data interpretation.

Based on the previously obtained results, we developed an MFC approach for MRD detection in patients with BCP-ALL treated with CD19-directed immunotherapy.

### 2.2. Protocol Testing: Patients and Samples

The developed approach was tested on 433 BM samples obtained from 65 children and young adults with R/R BCP-ALL (age: median, 9.5 years; range, 0.9–20.7 years) who received CD19- and CD19/CD22-directed CAR-T cells (NCT03467256 and NCT04499573, respectively) (Appendix A). According to the protocol, MRD detection was performed on days 14 and 28 and 2, 3, 4, 6, 8, 10, 12 and 24 months after the CAR-T infusion. The applicability of the developed algorithm was tested by a direct qualitative comparison with the MRD results obtained through RQ-PCR for FGTs (FGT-MRD) and through the NGS detection of *IG/TR* gene rearrangements (NGS-MRD). MRD results were compared qualitatively, as it is impossible to perform a direct comparison of the residual leukemic cell proportion measured by MFC and the relative MRD reduction measured by molecular techniques. The concordance rate was estimated as the proportion of concordant (both MRD-positive and MRD-negative) samples among all samples examined. Additionally, diagnostic sensitivity and specificity for MFC-MRD detection (using NGS-MRD as a reference) were calculated according to the conventional procedure [25]. Samples with a high leukemic burden (greater than 5% of residual leukemia by MFC) were excluded from the analysis. MFC-MRD and NGS-MRD results were compared in 338 samples, whereas the comparison of the MFC-MRD and FGT-MRD results was performed in 128 samples.

### 2.3. MRD Monitoring by MFC

MFC-MRD detection was performed with an 11-color panel (10 antibodies and DNA-stain Syto41, Thermo Fisher Scientific, Waltham, MA, USA) generally, according to the I-BFM-FLOW network guidelines [26], but with an adjustment for possible CD19 loss. Bone marrow aspirates were stained according to the manufacturer’s instructions, with appropriate amounts of directly conjugated monoclonal antibodies (MoABs) used for the sample cellularity. Erythrocytes were lysed with FACS Lysing solution (Beckton Dickinson, BD, San Jose, CA, USA). An IntraSure kit (BD) was used for permeabilization. Samples were acquired using a 3-laser CytoFLEX flow cytometer (Beckman Coulter, BC, Indianapolis, IN, US). EuroFlow guidelines for machine performance monitoring were used [27]. CytoFLEX Daily QC Fluorospheres (BC) were used for daily cytometer optimization. At least 300,000 nucleated cells were studied. MFC data were analyzed with Kaluza 2.1 software (BC). Lymphoblasts were considered leukemic if they represented a distinct population with leukemia-associated phenotypes and lymphoid light-scatter parameters. As per the I-BFM-FLOW network guidelines [26] and in accordance with the EuroFlow standardized approach [28], we defined a minimum of ten clustered leukemic lymphoblasts to consider a sample as MRD positive (lower limit of detection—LOD). The proportion of the leukemic cell population was presented as the percentage of Syto41-positive cells.

### 2.4. Cytogenetics and Molecular Genetics (MRD Monitoring by FGT-PCR)

Initial genetic diagnostics included conventional karyotyping and fluorescence in situ hybridization (FISH). Briefly, BM aspirates were cultivated overnight without mitogenic stimulation, and G-banding was performed as previously described [29]. Karyotypes were analyzed according to the International Cytogenomic Nomenclature, ICSN 2020 [30]. Samples were subjected to FISH with the corresponding probes (Appendix A) and RT-PCR [31] for AL-associated recurrent chromosomal aberrations.

Fusion transcript detection for MRD monitoring was carried out by quantitative reverse transcription-polymerase chain reaction (RT-PCR) in monoplex systems. Total DNA and RNA were simultaneously extracted from the BM samples using the InnuPrep DNA/RNA Mini Kit (Analytik Jena AG, Jena, Germany). For ALLs harboring t(12;21)(p13;q22)/*ETV6::RUNX1*, t(1;19)(q23;p13)/*TCF3::PBX1* and t(9;22)(q34;q11)/*BCR::ABL1,* the Europe Against Cancer primer-probe systems were used [31]. *KMT2A*-rearranged samples were subjected to NGS with a custom panel for the whole *KMT2A* gene [32]. The respective fusion transcripts were monitored with corresponding primer-probe systems [33]. Samples with any detectable FGT expression were considered MRD-positive. Quantification was performed with the Qiagen Ipsogen standards (Qiagen, Hilden, Germany), and ABL was used as a control gene [34]. t(17;19)(q22;p13)/*TCF3::HLF* samples were Sanger sequenced after a long-range genomic PCR. For *KMT2A*::EPS15 and *TCF3::HLF*, patient-specific primers were designed (Appendix A).

### 2.5. MRD Monitoring Using NGS of Specific IG/TR Gene Rearrangements

Genomic DNA was isolated from BM aspirates using a QIAamp DNA Mini Kit (Qiagen) according to the manufacturer’s protocol. The initial detection of *IG/TR* gene rearrangements was performed with an MRD detection kit (MiLaboratories LLC) according to the manufacturer’s protocol. The obtained amplicons were sequenced on an Illumina MiSeq (PE 150+150), generating approximately 10,000 sequencing reads per sample. MiXCR software [35] was used for clonotype assembly from raw sequencing data. Clonotypes with a frequency of 5% or higher were considered leukemic clone-specific rearrangements.

MRD monitoring was performed by next-generation sequencing of identified clonal *IG/TR* rearrangements, as described previously [36]. Approximately 5.5 micrograms of genomic DNA isolated from BM on days 28, 60, 90 and 120 was split into 32 aliquots: 16 aliquots of 300 ng each (equivalent to 50,000 cells), 8 aliquots of 30 ng each (equivalent to 5,000 cells) and 8 aliquots of 3 ng each (equivalent to 500 cells). Each aliquot was amplified independently with a combination of primers corresponding to specific V (or D in the case of incomplete DJ rearrangements of the TCR beta, delta or IGH locus) and J segments characteristic of the initially identified leukemic clone-specific rearrangement. Each first 25 µL PCR contained 1× Taq Turbo Buffer, 1 U of HS-Taq polymerase, 200 µM of each dNTP (all Evrogen, Russia) and 0.2 µM of each V (or D) and J gene-specific primer. The amplification profile was 95 °C for 2 min, followed by 10 cycles of 20 s at 95 °C, 20 s at 58 °C, and 40 s at 72 °C and 15 cycles of 20 s at 95 °C and 60 s at 72 °C. Then, a 2 µL aliquot of the first PCR was transferred to the second (indexing) 25 µL PCR containing 1× Taq Turbo Buffer, 0.5 U of HS-Taq polymerase, 200 µM of each dNTP (all Evrogen, Moscow, Russia) and 0.2 µM of each Illumina Nextera XT indexing primer. The amplification profile was 95 °C for 2 min, followed by 20 cycles of 20 s at 95 °C, 20 s at 60 °C and 40 s at 72 °C. Equal volumes of the 32 final libraries were pooled, purified with AmPure XP beads according to the manufacturer’s protocol and sequenced on an Illumina MiSeq (PE 150+150), generating approximately 1,000,000 sequencing reads per sample (i.e., per 32 libraries, mixed). MiXCR software was used for the clonotype assembly from raw sequencing data. Poisson statistics with the λ estimator λ = −ln(1 − p) (where λ is the expected portion of target molecules and p is the observed portion of target molecules) [37] were used to calculate the initial concentration of each rearrangement. In the case of multiple leukemic clone-specific rearrangements, the one with the highest concentration was used as the MRD value.

## 3. Results

### 3.1. Algorithm Description

An eleven-color antibody panel was developed based on the obtained data. It consisted of antibodies for the detection of the CD19, CD10, CD45, CD38, CD34, CD58, CD20, CD22 and CD24 surface expression, as well as the cytoplasmic expression of CD79a. The antibody clones and fluorochromes used are shown in Table 1. CD58, CD10, CD34, CD38, CD19, CD20 and CD45 were combined into custom-designed, ready-to-use lyophilized BD™ Lyotubes (BD). CD22, CD24 and CD79a were used as drop-in antibodies. This composition of the panel construction aimed to unify the use of the premanufactured tube for all patients with BCP-ALL, irrespective of the immunotherapy applied. Syto41 is also a drop-in reagent and helps to separate nucleated cells from debris [11].

The general algorithm of the residual tumor cell search is shown in Figure 1. The combination of CD22 and iCD79a was chosen as the main substitution for CD19 (Appendix A). According to the expression of these antigens by tumor cells before the start of immunotherapy, we suggest either the consequent gating of the B-cell compartment using CD22 and then iCD79a (if both antigens are expressed strongly), or the application of one single antigen if the second antigen is not totally expressed. If neither CD22 nor iCD79a is totally expressed before immunotherapy initiation, the primary B-cell gating should be performed using CD10 or CD24. As these two antigens also detect mature neutrophils, additional purification of the B-lineage gate could be needed: CD45 expression and SSC values can be useful for the exclusion of granulocytes. Nevertheless, according to the described algorithm of the B-lineage compartment gating, CD24 or CD10 are solely used in extremely rare cases (less than 2% of samples) [20]. 

Usually, two approaches are applied for searching for residual leukemic cells inside the B-lineage compartment [38,39]: the identification of cells with the initially detected LAIP and cells that are different in terms of the antigen expression profile from the normal patterns of B-cell development (the so-called “different from normal” (DFN) technique) (Figure 1). It is known that the expression of all antigens, not only CD19, can be changed in both directions by CD19 immunotherapeutic targeting, especially in patients with R/R disease, and the DFN method of MRD investigation should be considered more appropriate. At the same time, several pitfalls in the application of the DFN approach also exist. This method is based on the understanding of patterns of normal B-cell differentiation and the presence of “empty spaces” on the dot plots, in which leukemic blasts can be found. It is necessary to realize that the placement of “empty spaces” on the dot plots completely depends on the algorithm of primary B-lineage gating, because different stages of B-lineage maturation as well as cells of a non-B-cell origin can be included in such a primary gate (Table 2). The maximal range of normal cell populations will appear in the rough B-cell region if only CD22 is used for gating: basophils [40,41] and plasmacytoid dendritic cells (PDCs) [42] with dim CD22 positivity will be gated together with all stages of BCP maturation and mature B-lymphocytes (Appendix A). Only cells of a B-cell origin will be gated if iCD79a is solely applied, but plasma cells will also be seen on the dot plots (Appendix A). The consecutive application of both markers allows us to see only normal BCPs (both CD19-positive and CD19-negative) and mature lymphocytes as the normal background of the MRD search (Appendix A). The narrow list of normal cell populations can be gated by CD24 (Appendix A), as plasma cells and CD19-negative BCPs do not express this antigen, whereas CD10-based primary gating will include only BCPs irrespective of their CD19 expression. Of note, the relative expansion of normal CD19-negative BCPs after CD19-directed immunotherapy can make these cells highly visible, and their misinterpretation as being a part of the development of the CD19-negative relapse should be avoided.

Lineage conversion, which became not an extremely rare event after the wide implementation of immunotherapy, should also be considered during MRD investigation [24]. With the aim of the timely diagnosis of such a form of tumor escape, our approach includes the obligatory examination of CD45, CD38, CD34 and CD24 expression vs. side-scatter (SSC) to find any suspicious cell populations among all nucleated cells (Figure 1).

### 3.2. Protocol Testing

In total, MFC-MRD and NGS-MRD results were compared in 338 BM samples, whereas MFC-MRD and FGT-MRD data comparisons were performed in 128 samples. Overall, 82.8% of the samples displayed qualitative concordance between MFC-MRD and NGS-MRD, and in 89.8% of the samples, the MFC-MRD and FGT-MRD results were concordant (Table 3). In 12 of 27 cases with NGS-MRD positivity and MFC-MRD negativity, MFC-MRD appeared at the next MRD monitoring time point. The same was seen in three of the six MFC-MRD-negative and FGT-MRD-positive cases. Similarly, discordant MFC-MRD positivity preceded the appearance of MRD measured by molecular techniques.

Samples from patients with *KMT2A* gene rearrangements were analyzed separately, as this subgroup of BCP-ALL is known for immunophenotypic peculiarities that can hamper MFC-MRD detection [43,44], especially if CD19 is lost. MFC-MRD and NGS-MRD were compared in 37 samples with a concordance rate of 86.5%, whereas for the MFC-MRD vs. FGT-MRD comparison, this rate was 98.1% (*n* = 54) (Table 3).

Patients who relapsed or displayed MRD reappearance after CAR-T therapy were also analyzed separately (Table 4). In samples obtained before the reappearance of CD19-positive blasts, the MFC-MRD and NGS-MRD results were concordant in 78.8% (52 of 66) of samples, whereas the MFC-MRD and FGT-MRD results were concordant in 90.3% (28 of 31) of samples. With the use of NGS-MRD as a reference method, the diagnostic sensitivity of the MFC-MRD approach was calculated as 45.5%, whereas diagnostic specificity was 98.7%. For patients with a CD19-negative recurrence, the concordance rates of the MFC-MRD results were similar: 85.1% for NGS-MRD (63 of 74 samples evaluated) and 88.5% for FGT-MRD (23 of 26 samples evaluated).

Moreover, we noted a difference in the comparability of the MFC-MRD results with different molecular techniques obtained early after CAR-T infusion (during the first two months): the concordance with the NGS-MRD results was 73.1%, whereas the concordance with the FGT-MRD results was 97.0% (Table 5). At later time points, the comparability of the MFC-MRD data with both NGS-MRD and FGT-MRD data was similar (85.3% and 87.1%, respectively, Table 5).

In 49 samples that were found to be MFC-MRD-positive, the median value of MFC-MRD was 0.135% (range 0.002%–4.771%). Of note, 13 of these samples displayed an MFC-MRD positivity below 0.01%, showing applicability of the conventional LOD threshold of the MFC-MRD measurement also for an approach based on the CD19-independent gating strategy.

## 4. Discussion

The wide implementation of CD19-directed therapy in the treatment algorithms for both primary and R/R BCP-ALL [5,6,7,45,46,47] significantly complicated the rather routine procedure of MFC-MRD monitoring. More sensitive molecular techniques seem to be more applicable after immunotherapy [10,48], as their reproducibility is not directly linked with the expression of therapeutic targets. Nevertheless, MFC still holds serious advantages in comparison with either RQ-PCR or even NGS [3,10]. In addition to their relatively high cost, which seriously limits their applicability, molecular techniques do not provide some clinically relevant information, such as antigen expression profiles and possible therapeutic targets. In addition, the reappearance of MRD after immunotherapy in nearly all situations calls for changing the therapeutic approach, including possible new immunotherapeutic drug applications [49]. Thus, MFC-MRD monitoring retains its significance as the crucial method for assessing therapeutic efficacy [10]. Therefore, it is necessary to develop a reliable MFC-based approach for MRD monitoring in patients who undergo CD19 targeting.

Based on the published data from other groups [15,16,17,50] and our consecutive studies [18,19,20,22,23,24], we have developed, validated and implemented an 11-color single-tube approach for MFC-MRD detection in routine practice. Ready-to-use premanufactured tubes containing the majority of used antibodies can seriously improve the reliability of the assay by minimizing possible pipetting errors and providing highly stable cytometric data, which can be analyzed not only manually but also with programmed approaches [28,51,52]. Either custom-designed tubes, as in our study, or commercially available kits [53] can be used with similar success. The application of such a multicolor antibody panel that includes several antigens suitable for primary B-cell gating allows for choosing the optimal gating strategy not only based on the antigen expression prior to the start of immunotherapy, but also considering possible treatment-related immunophenotypical changes. Only a difference in the incidence of CD19 loss was observed between patients, who were treated with blinatumomab or CAR-T [18,19]. Changes of expression in other antigens [18,19], as well as normal background modulation [22], did not differ between these two types of CD19 targeting. At the same time, the incidence of CD19 negativity after blinatumomab is high enough [8,18] to consider CD19-based gating algorithms inappropriate. Therefore, we suggest using the same CD19-independent approach without respect for the type of CD19-directed immunotherapy.

We suggest that the DFN approach is more suitable for MFC-MRD monitoring after CD19-directed immunotherapy, and then for searching for LAIPs, considering that patients could be heavily pretreated (especially R/R patients) and that the tumor phenotype could also be changed by immunotherapy itself. The application of the DFN method requires a precise understanding of what cells are included in the primary B-cell gate when different antigens and their combinations are used for this gating. CD19-negative normal BCPs, basophils and PDCs could be the most frequently misinterpreted leukemic blast cells because the “classic” MFC-MRD approach starts from CD19-positive cell gating; therefore, the majority of researchers are not used to the mentioned cells within the primary B-cell gate. In the case of any doubts with regard to the leukemic origin of the found cells, we recommend the purification of the suspicious populations via flow cell sorting with consecutive molecular studies, i.e., a chimerism evaluation in transplanted patients, the detection of *IG/TR* gene clonality or leukemia-specific gene fusions [54,55]. All of these studies could be relevant to prove or to rule out suspect MFC-MRD results.

The designed approach was tested on a relatively large set of samples obtained from patients treated with CAR-T therapy. The comparison of the MFC-MRD data with results obtained by two different molecular techniques (FGT-MRD and NGS-MRD) was chosen as the method of technology validation. We obtained an overall concordance rate of 82.8% between MFC-MRD and NGS-MRD. Similar comparability was also achieved between MFC-MRD and FGT-MRD (89.8%). These levels of concordance are in full agreement with previously published data for both molecular techniques [28,56,57,58]. Moreover, this level of comparability is more or less common for all comparisons between MFC and other MRD techniques [59,60]. At the same time, the known fact that MFC-MRD is less sensitive than FGT-MRD and NGS-MRD must be acknowledged. This was also confirmed in our study because all detected discrepancies were found to be the result of either a “faster” MRD clearance detected by MFC at the beginning of treatment or the earlier detection of MRD reappearance before relapse. Both scenarios are mainly caused by differences in the sensitivity of the methods. Interestingly, neither the presence of *KMT2A*-r nor the further loss of CD19 by leukemic cells led to discrepant results of MFC-MRD detection and other techniques. Of note, the same comparability of MFC-MRD and FGT-MRD was obtained in a previous study of infants with BCP-ALL and *KMT2A*-r, whereas FGT-MRD data in that study were obtained from another molecular laboratory [58].

Our MFC-MRD approach differs from methods suggested by other groups. S. Cherian et al. introduced the possible use of CD22 and CD24 instead of CD19 after immunotherapy [15]. While acknowledging the high potential of these markers, we do not think that their application is sensitive enough in specific patient subgroups, i.e., for example, *KMT2A*-r [43,44], which is relatively enriched among R/R BCP-ALL patients. Moreover, the application of CD22 and CD24 requires additional staining for myeloid antigens and, if incorporated within current eight-color panels, increases the number of tubes required [50]. The use of iCD79a as a solid antigen for the B-cell compartment [16] also has limitations, because intracellular staining in samples from heavily pretreated patients could lead to equivocal distribution between positive and negative cells, which can be overcome by the use of other B-cell markers in combination with iCD79a, as we suggest. The currently known EuroFlow approach [17] is a modification of their conventional panels and gating algorithms. This modification is mainly based on CD10 and CD34 expression, which is sometimes not total even in diagnostic samples. Nevertheless, it was announced that the new MFC-MRD methodology for after CD19-directed immunotherapy designed by the EuroFlow consortium will be published soon [17].

## 5. Conclusions

In conclusion, we developed an 11-color single-tube MFC approach for patients with BCP-ALL who undergo CD19-directed immunotherapy. The suggested gating algorithm and new knowledge of possible changes in both leukemic and normal cell properties enable the approach to overcome the possible loss of CD19, which was considered the main B-lineage gating antigen for three previous decades. The results of the method testing show that our technology retains the high effectiveness and reproducibility of MFC as the method of MRD monitoring even in the case of CD19 targeting, and the MRD evaluation results remain reliable, as shown for patients treated with chemotherapy only.

## Figures and Tables

**Figure 1 cancers-14-05445-f001:**
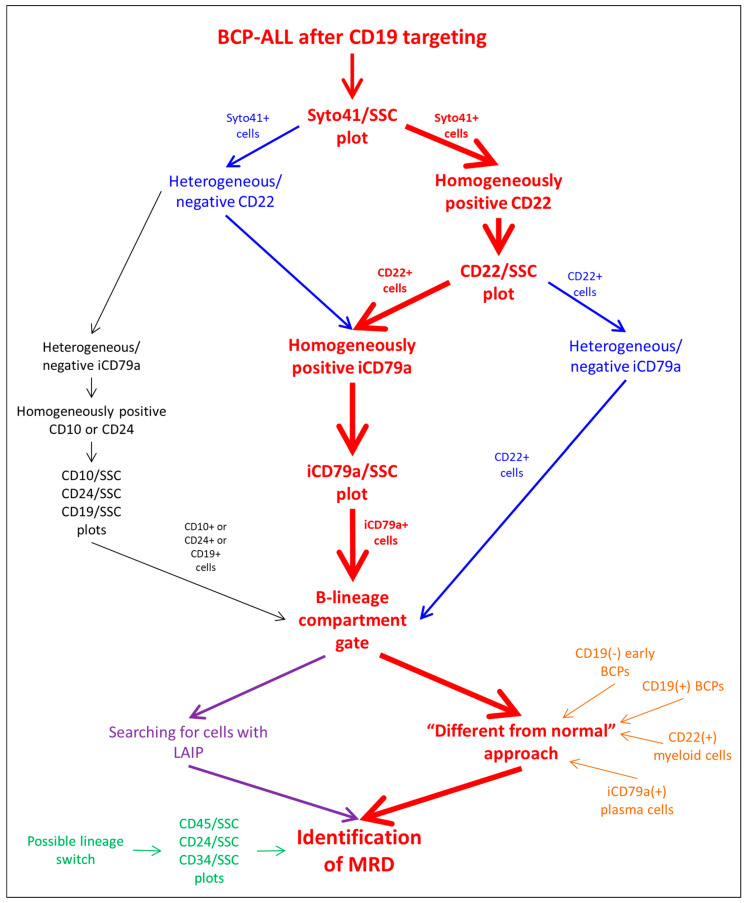
Analysis algorithm of BCP-ALL MRD detection after CD19-directed therapy.

**Table 1 cancers-14-05445-t001:** Eleven-color antibody panel for MFC-MRD in BCP-ALL patients after CD19-directed therapy with information about clones, fluorochromes and manufacturers.

Fluorochrome	FITC	PE	PE-CF594	PerCP-Cy5.5	PE-Cy7	APC	A700	APC-Cy7		BV510	BV768
**Antibody**	CD58	CD22	CD10	CD20	CD19	iCD79a	CD34	CD45	**SYTO41**	CD38	CD24
**Clone**	1C3 (AICD58.6)	S-HCL-1	HI10a	L27	SJ25C1	HM47	8G12	2D1		HIT2	ML5

**Table 2 cancers-14-05445-t002:** Number of normal cell populations in the B-cell gate according to the gating strategy.

B-Cell Gating Type	Normal BM Cell Populations
CD19-based	CD19(+) BCPs, plasma cells, mature B cells
CD22+iCD79a-based	CD19(+) BCPs, CD19(−) BCPs, mature B cells
CD22-based	CD19(+) BCPs, CD19(−) BCPs, mature B cells, basophils, plasmacytoid dendritic cells
iCD79a-based	CD19(+) BCPs, CD19(−) BCPs, plasma cells, mature B cells
CD10-based	CD19(+) BCPs, CD19(−) BCPs
CD24-based	CD19(+) BCPs, mature B cells

**Table 3 cancers-14-05445-t003:** Qualitative concordance between MFC-MRD and NGS-/FGT-MRD results in all samples and in samples with *KMT2A*-rearranged BCP-ALLs.

**All samples**		NGS-MRD	
		Positive	Negative
MFC-MRD	Positive	46	3
	Negative	55	234
Concordance: 82.8%			
**All samples**		FGT-MRD	
		Positive	Negative
MFC-MRD	Positive	12	3
	Negative	10	103
Concordance: 89.8%			
** *KMT2A* ** **-rearranged samples**		NGS-MRD	
		Positive	Negative
MFC-MRD	Positive	3	0
	Negative	5	29
Concordance: 86.5%			
** *KMT2A* ** **-rearranged samples**		FGT-MRD	
		Positive	Negative
MFC-MRD	Positive	3	0
	Negative	1	50
Concordance: 98.1%			

**Table 4 cancers-14-05445-t004:** Qualitative concordance between MFC-MRD and NGS-/FGT-MRD results according to CD19 expression on leukemic cells after CD19 CAR-T or bispecific CD19/CD22 CAR-T therapy.

**CD19(+) blasts after CAR-T therapy**		NGS-MRD	
		Positive	Negative
MFC-MRD	Positive	11	0
	Negative	14	41
Concordance: 78.8%			
**CD19(+) blasts after CAR-T therapy**		FGT-MRD	
		Positive	Negative
MFC-MRD	Positive	1	2
	Negative	1	27
Concordance: 90.3%			
**CD19(−) blasts after CAR-T therapy**		NGS-MRD	
		Positive	Negative
MFC-MRD	Positive	21	2
	Negative	9	42
Concordance: 85.1%			
**CD19(−) blasts after CAR-T therapy**		FGT-MRD	
		Positive	Negative
MFC-MRD	Positive	4	1
	Negative	2	19
Concordance: 88.5%			

**Table 5 cancers-14-05445-t005:** Qualitative concordance between MFC-MRD and NGS-/FGT-MRD results according to time point after CD19 CAR-T or CD19/CD22 CAR-T infusion.

**Less than 2 months after infusion**		NGS-MRD	
		Positive	Negative
MFC-MRD	Positive	4	1
	Negative	17	45
Concordance: 73.1%			
**Less than 2 months after infusion**		FGT-MRD	
		Positive	Negative
MFC-MRD	Positive	2	0
	Negative	1	30
Concordance: 97.0%			
**2 months or more after infusion**		NGS-MRD	
		Positive	Negative
MFC-MRD	Positive	22	1
	Negative	27	140
Concordance: 85.3%			
**2 months or more after infusion**		FGT-MRD	
		Positive	Negative
MFC-MRD	Positive	3	2
	Negative	6	51
Concordance: 87.1%			

## Data Availability

The datasets generated during and/or analyzed during the current study are available from the corresponding author on reasonable request.

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
