# Peer review of "Reliable Flow-Cytometric Approach for Minimal Residual Disease Monitoring in Patients with B-Cell Precursor Acute Lymphoblastic Leukemia after CD19-Targeted Therapy"

_cancers, 2022, doi:10.3390/cancers14215445_

Round 1

Reviewer 1 Report

In their manuscript, Mikhailova et al. present multicolor flow cytometry monitoring for residual disease in pediatric BCP-Acute Lymphoblastic Leukemia patient samples post T cell engaging anti CD19 therapy. The DFN approach highlighted by the authors in the study suggests a potential MFC based analysis to detect MFC-MRD in the absence of CD19 expression. Overall, the authors have proposed an effective method to monitor MFC-MRD; however, a few concerns need to be addressed.

1-    The authors should provide the age category of the patients in the study.

2-    The authors should include a neutrophil exclusion strategy In CD22-CD24cell population.

Author Response

Referee #1 (Comments to the Author):

In their manuscript, Mikhailova et al. present multicolor flow cytometry monitoring for residual disease in pediatric BCP-Acute Lymphoblastic Leukemia patient samples post T cell engaging anti CD19 therapy. The DFN approach highlighted by the authors in the study suggests a potential MFC based analysis to detect MFC-MRD in the absence of CD19 expression. Overall, the authors have proposed an effective method to monitor MFC-MRD; however, a few concerns need to be addressed.

Response to Referee 1

Comment 1.

The authors should provide the age category of the patients in the study

Response.

We have added this information in lines 138-139.

Comment 2. 

The authors should include a neutrophil exclusion strategy In CD22-CD24+ cell population.

Response.

We thank the reviewer for this comment. In our gating algorithm CD24 as a single antigen for the B-lineage compartment gating is applied in a very small proportion of patients: below 2% of cases. Moreover, in these cases CD24 could be used together with CD10 and, if applicable, with CD19 as well. Therefore, the necessity of neutrophil exclusion is limited to the very few cases. This point is highlighted in the Discussion section (lines 372-374). In addition, we have now added the statement of possible use of CD45 expression and SSC parameter values for such neutrophil exclusion, if necessary (lines 241-245)

Reviewer 2 Report

In this manuscript, Authors describe an antibody panel and data analysis algorithm for multicolor flow 16 cytometry (MFC), which is a reliable method for MRD detection in patients with B-cell precursor acute lymphoblastic leukemia treated with CD19-directed therapy. This paper is well written and valuable. The limited of this study is a small group of children (65pts). I suggest to improve quality of Figure 1.

Author Response

Referee #2 (Comments to the Author):

In this manuscript, Authors describe an antibody panel and data analysis algorithm for multicolor flow 16 cytometry (MFC), which is a reliable method for MRD detection in patients with B-cell precursor acute lymphoblastic leukemia treated with CD19-directed therapy. This paper is well written and valuable.

Response to Referee 2

Comment 1.

The limited of this study is a small group of children (65pts).

Response.

Actually, the approach was constructed on the basis of studying of few hundreds of patients, both for the initial antigens expression and for immunophenotypic changes of leukemic and normal cells under immunotherapy. Samples from 65 patients were used only for the approach testing, because there were patients, who underwent more or less uniform treatment and routine consecutive MRD measurement with 2-3 methods. Due to significant immunophenotypic changes observed, in several patients the algorithms of cytometric data analysis varied between samples obtained at different time-points. That is why we think, that the cohort of tested samples and patients can be considered enough informative for the testing of our method applicability.

Comment 2.

I suggest to improve quality of Figure 1. 

Response.

Now we provide the Figure in higher resolution

Reviewer 3 Report

In this ms., Mikhilova and colleagues described a new panel of antibodies developed to detect MRD after CAR-T cell infusion therapy. It is mainly based on the gating of CD22 and iCD79a in order to detect and to diagnose CD19-negative ALLs arising after CAR-T cell therapy.

The paper is very well written and the data are presented and concise. The method is not novel and has been described in the literature before. Hence, there are many issues, which should be addressed prior publication. 

Major comments

-       The authors should not only give the concordance as quality control between NGS-MRD and MFC-MRD, and between FGT-MRD and MFC-MRD but also the sensitivity and specificity of their new developed method, f.e. considering the NGS-MRD method as gold standard.

-       The authors reported a concordance of 73% in specific time points  (@ less than 2 months after infusion”) the lack of concordance is mainly due to false negative in the MFC-MRD technique compared to the NGS-MRD. Therefore, it the new developed method does not seem to be very sensitive to detect MRD after CAR-T therapy.  Do the authors have used their new developed panel of antibodies in patients with ALL during chemotherapy treatment? Have the authos compared their new developed antibody with the standard CD19-based panel antibody to detect MRD?

-       The authors should also describe which is the limit of detection of their new approach compared to a CD19-based MFC-MRD approach.

-       The authors suggested, that their approach may be also used for other immunotherapies as blinatumumab. However, the authors do not show any data related to blinatumumab. Can the authors show any data using their approach to patients under blinatumumab treatment?

Minor comments

-       Table S1: Please, correct kariotype by karyotype.

Author Response

Referee #3 (Comments to the Author):

In this ms., Mikhailova and colleagues described a new panel of antibodies developed to detect MRD after CAR-T cell infusion therapy. It is mainly based on the gating of CD22 and iCD79a in order to detect and to diagnose CD19-negative ALLs arising after CAR-T cell therapy.

The paper is very well written and the data are presented and concise. The method is not novel and has been described in the literature before. Hence, there are many issues, which should be addressed prior publication.

Response to Referee 3

Comment 1.

The authors should not only give the concordance as quality control between NGS-MRD and MFC-MRD, and between FGT-MRD and MFC-MRD but also the sensitivity and specificity of their new developed method, f.e. considering the NGS-MRD method as gold standard.

Response.

We have added this information in lines 293-295. We think that the diagnostic sensitivity (as well as overall concordance) is highly dependent on the set of samples studied. In our study we tested our approach on the unselected set of consecutive samples, therefore we have more or less “real world” results. The low diagnostic sensitivity here is mainly caused by the lower analytical sensitivity of MFC, comparing to NGS: these differences in MRD-positivity were mainly observed at the beginning of treatment and before relapse (if occurred). These points are discussed in lines 358-363.

Comment 2. 

The authors reported a concordance of 73% in specific time points  (@ less than 2 months after infusion”) the lack of concordance is mainly due to false negative in the MFC-MRD technique compared to the NGS-MRD. Therefore, it the new developed method does not seem to be very sensitive to detect MRD after CAR-T therapy.  Do the authors have used their new developed panel of antibodies in patients with ALL during chemotherapy treatment? Have the authors compared their new developed antibody with the standard CD19-based panel antibody to detect MRD?

Response.

We thank the reviewer for this comment. Actually, we did not aim to develop MFC approach, which has equal sensitivity to NGS. We consider this impossible in acute leukemia now. Even using so-called “next generation flow”, the really high sensitivity is currently achieved in routine practice only for multiple myeloma monitoring. For ALL it is still lower than NGS can achieve. Nevertheless, in the Era of immunotherapy it is impossible to use only the most sensitive NGS technology, because in the case of MRD reappearance it is vital to assess the antigen expression profile in order to choose appropriate immunotherapeutic drug. That is why we think, that it is possible to admit the “faster” MRD clearance by MFC, than by NGS as the most crucial meaning of MRD in the context of CAR-T therapy is detection of MRD reappearance after MRD-negative remission. The clinical value of MFC-MRD-negativity after blinatumomab application even in a case of FGT-MRD-positivity was shown previously on the model of infants with KMT2A-r ALL (A. Popov et al, BJH, 2021, A. Popov et al, BJH, 2022). These points are discussed in lines 309-321. Therefore, although we have to admit, that MFC in any case cannot be as sensitive, as NGS, we have to develop MFC methodology, which will allow us to detect MRD in a case of possible CD19 loss with the same effectiveness that we have without CD19 targeting. So, our primary aim was to develop methodology (antibody panel, algorithm of analysis, etc) which will allow us to have the same efficacy either with or without immunotherapy applied. That is why we are satisfied that we obtained the same level and direction of discrepancy with NGS, as MFC typically has, when only chemotherapy is applied. These points are discussed in lines 358-363. Generally, we had an idea to add additional B-lineage antigens to conventional MFC-MRD panel, while the key role of CD19 is substituted by other antigens in the analysis algorithm. At the same time, if only chemotherapy is used, the effectiveness of our panel will be the same by definition, as the panel allows conventional CD19-based gating as well. Moreover, the design of pre-manufactured tubes (both commercially available and custom) contains only conventional antibodies, typical for the groups, participating in I-BFM Flow network (CD19, CD10, CD20, CD45, CD34, CD38, CD58), while the additional B-lineage antibodies are suggested as the drop-in reagents. This flexibility allows using the same tube for all cases of BCP-ALL, although for patients who underwent CD19-directed treatment, additional markers could be added. These points are discussed in lines 230-234.

Comment 3. 

The authors should also describe which is the limit of detection of their new approach compared to a CD19-based MFC-MRD approach.

Response.

We thank the reviewer for this comment. We have added the definition of LOD, that we used according to the guidelines of I-BFM-FLOW network, which is in accordance with EuroFlow standardized approach (lines 166-170). Actually, the ability of detection of low cell quantities did not suffer from gating algorithm changes. First, as described in the answer to the previous comment, we had an idea to add additional B-lineage antigens to conventional MFC-MRD panel, while the key role of CD19 is substituted by other antigens in the analysis algorithm. Moreover, as it is shown in lines 255-269, Table 2 and Figures S2-S5, using the suggested panel of antibodies, it is possible to dissect clearly all normal cell populations that are included in the analysis in each way of B-lineage compartment gating. Therefore, searching for the residual leukemic cells in these ways of gating has the same effectiveness with the conventional CD19-based gating strategy. We added the MFC-MRD levels distribution in MFC-MRD-positive samples from the test cohort (lines 303-306), showing the substantial proportion of samples with very low MFC-MRD values, that confirms applicability of the conventional LOD values also to the designed approach.

Comment 4. 

The authors suggested, that their approach may be also used for other immunotherapies as blinatumumab. However, the authors do not show any data related to blinatumumab. Can the authors show any data using their approach to patients under blinatumumab treatment?

Response.

We thank the reviewer for this comment. We have observed no significant differences in changes of normal BM background and immunophenotype of leukemic cells between patients, who were treated with blinatumomab or CAR-T. Only the incidence of CD19-negativity was higher in a case of CAR-T. Nevertheless, incidence of CD19 loss after blinatumomab was enough high to consider CD19-based gating inappropriate. Therefore, we suggest using the same approach without respect to the type of applied immunotherapy. This statement is added in lines 332-337. In our country, historically MFC is used as the main method of MRD monitoring in ALL. Only for the CAR-T trials, NGS was implemented as the comparative technology. Therefore, it was impossible for us to provide comparison between MFC and NGS in other patients. We can say, that FGT-MRD was compared to MFC-MRD also in infants with KMT2A-r ALL (A. Popov et al, BJH, 2022) with the same rate of concordance. This point is mentioned in the lines 364-367.

Comment 5. 

Table S1: Please, correct kariotype by karyotype.

Response.

Corrected